# Chain-of-Questions Training with Latent Answers for Robust Multistep Question Answering

**Wang Zhu**     **Jesse Thomason**     **Robin Jia**
University of Southern California, Los Angeles, CA, USA
{wangzhu, jessetho, robinjia}@usc.edu

## Abstract

We propose Chain-of-Questions, a framework that trains a model to robustly answer multistep questions by generating and answering sub-questions. We obtain supervision for sub-questions from human-annotated question decomposition meaning representation (QDMR), but QDMR does not include annotated answers to sub-questions. To overcome this technical challenge, we treat sub-answers as latent variables and infer them with a novel dynamic mixture of Hard-EM and MAPO. Chain-of-Questions is effective and robust, greatly outperforming strong neuro-symbolic methods by 9.0 F1 on a DROP contrast set and GPT-3.5 by 24.3 F1 on a HOTPOTQA adversarial set.

## 1 Introduction

Multistep question answering (QA) poses a reasoning challenge that current state-of-the-art QA models have not fully addressed. Strong fine-tuned QA models like UnifiedQA (Khashabi et al., 2020a) can achieve impressive results on various QA tasks through multitask training, but exhibit subpar performance on multistep reasoning. Moreover, because some multistep reasoning benchmarks contain annotation artifacts or reasoning shortcuts (Jiang and Bansal, 2019), dedicated models trained on these benchmarks often have much lower F1 performance on contrast sets (Gardner et al., 2020) and adversarial sets (Schlegel et al., 2021), indicating their lack of robustness.

Prior research has attempted to tackle this challenge with various question decomposition strategies to explicitly incorporate reasoning chains into the question answering process. However, as we show in our experiments, existing methods (Andor et al., 2019; Chen et al., 2020) that perform explicit reasoning steps still suffer from robustness issues. Moreover, multi-step reasoning methods are often engineered for a specific domain or type

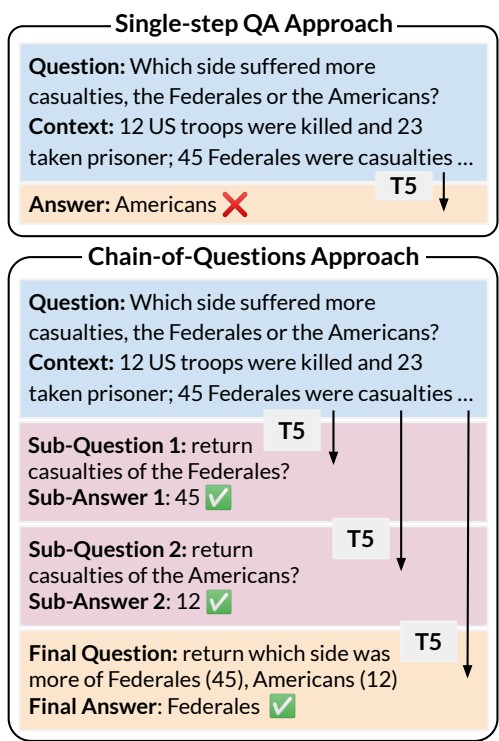

Figure 1: Single-step QA *vs*. Chain-of-Questions. We show that a single-model with sub-question generation and answering works better than single-step QA on questions that require multistep reasoning.

of multistep QA (Fu et al., 2021; Perez et al., 2020), and thus cannot be easily extended to other multistep QA settings. Prompting methods (Chen et al., 2020; Dua et al., 2022) have shown promise in generating multistep solutions to questions, but they require very large language models (LMs) as well as careful prompt engineering, and still lag behind fine-tuned methods (OpenAI, 2023).

To develop a robust multistep QA system, we propose a novel framework, Chain-of-Questions training with latent answers. Our framework trains a model to generate sub-questions and their corresponding sub-answers one at a time, as shown in Fig. 1, then aggregates those sub-answers to answer the original question. To define an appropriate

set of sub-questions, we use question decomposition meaning representation (QDMR), an existing dataset with human-annotated sub-questions for questions from multiple multistep QA benchmarks. While QDMR is helpful, it only contains annotated sub-questions, not sub-answers, which makes training a QA system to generate sub-answers technically challenging. We view the sub-answers in the intermediate steps as latent variables, and apply Hard-EM (Neal and Hinton, 1998) to optimize these latent variables during training. To further improve performance, we use a memory buffer to store trajectories with high F1 score, inspired by Memory-Augmented Policy Optimization (MAPO; Liang et al., 2018), previously used for semantic parsing. Because starting with MAPO alone does not converge well, we design a dynamic loss function that combines the Hard-EM and MAPO objectives for fast improvement at the beginning and better final convergence.

We conduct experiments on DROP (Dua et al., 2019), HOTPOTQA (Yang et al., 2018), and their contrast and adversarial sets to evaluate the performance of our proposed Chain-of-Questions framework. On the contrast set of DROP, Chain-of-Questions outperforms neuro-symbolic baselines by 9.0 on F1 score, and outperforms Chain-of-Thought on GPT-3.5 by 16.8 despite using a much smaller model (T5-Large, 770M parameters). On the adversarial set of HOTPOTQA, Chain-of-Questions outperforms Longformer by 5.5 on F1 score, and outperforms Chain-of-Thought on GPT-3.5 by 24.3. Our experimental results demonstrate that Chain-of-Questions successfully leverages existing QDMR annotations to train an effective and robust multistep QA model.

## 2 Background and Related Work

We introduce the multistep QA benchmarks we use, as well as other methods using question decomposition during training and prompting.

**Multistep QA Benchmarks.** We focus on two popular multistep QA benchmarks—DROP and HOTPOTQA. DROP (Dua et al., 2019) focuses on questions that require discrete and symbolic reasoning. Most of its questions require multiple steps of retrieval and numerical execution. HOTPOTQA (Yang et al., 2018) contains 2-hop questions over 10 paragraphs. Other work has constructed contrast and adversarial sets to evaluate the robustness of models trained on these

datasets. Gardner et al. (2020) created a contrast set for DROP (DROP-CS) by modifying test instances in ways that often change the correct answer. HOTPOTQA-ADV (Jiang and Bansal, 2019) adds adversarial paragraphs that do not change the correct answer but fool models that rely too heavily on reasoning shortcuts. We experiment on DROP, HOTPOTQA, and their robustness evaluation sets.

**Training with Question Decomposition.** Wolfson et al. (2020) introduce QDMR, a human-annotated question decomposition format and dataset. Since QDMR's for each question were annotated without looking at evidence passages, the QDMR dataset does not include sub-answers to any sub-questions within each QDMR. Subsequent work has used QDMR to help train QA models. TeaBReaC (Trivedi et al., 2022b) uses the QDMR decomposition graph to generate a large multistep QA dataset with synthetic contexts for pretraining. We show that TeaBReaC has complementary benefits with Chain-of-Questions, which adds explicit multistep reasoning at inference time. Guo et al. (2022b) train a model to generate QDMR's and provide them as context to a single-step QA model. Unlike this model, our model explicitly attempts to generate sub-answers of QDMR sub-questions. We compare with a single-step baseline similar to Guo et al. (2022b), and show that learning to generate sub-answers improves performance.

Other work on multistep QA, such as DecompRC (Min et al., 2019b), ONUS (Perez et al., 2020) and RERC (Fu et al., 2021), generate a decomposition with one model and the answers for the sub-questions with another model, although none of these use QDMR. These methods require a single-step QA model trained with other QA data, whereas our approach does not. Moreover, they rely on entity matching to decompose questions; such approaches do not naturally extend to tasks requiring forms of multistep reasoning that are not entity-centric, such as numerical reasoning.

Neuro-symbolic methods such as BERT-Calculator (Andor et al., 2019) and NeRd (Chen et al., 2020) generate functional programs for multistep numerical reasoning. However, they require the model to generate accurate programs in a single run, without observing the results of intermediate stages of computation. This process can make them more susceptible to learning simple reasoning shortcuts, compared with Chain-of-Questions.

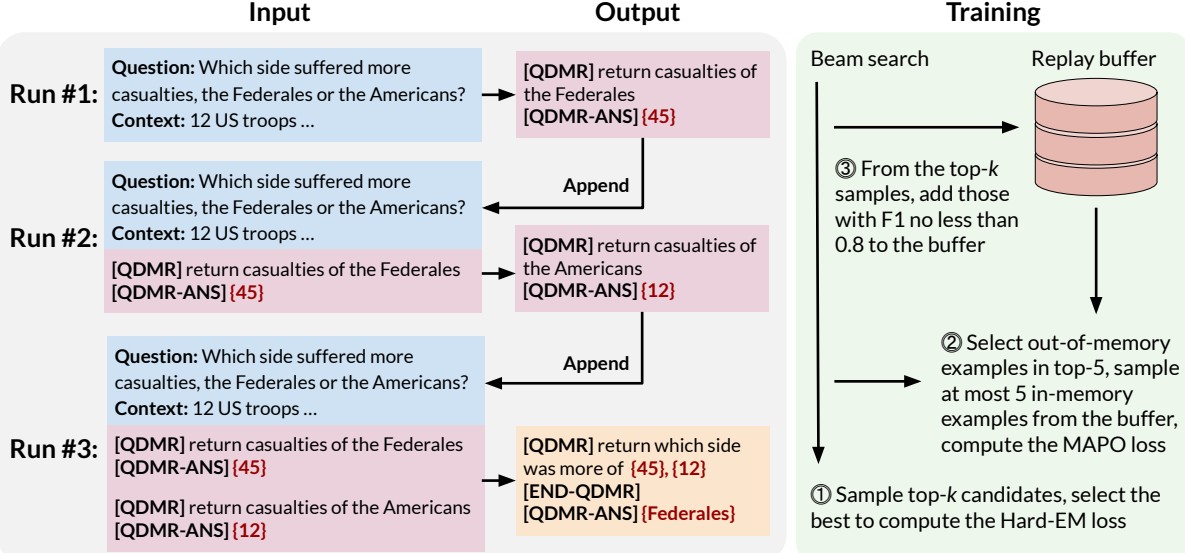

Figure 2: Chain-of-Questions framework and its training process with Hard-EM and MAPO. In the left panel, the blue box contains the input question and context, the pink box shows the intermediate sub-questions and sub-answers, the orange box is the final sub-question and final answer. Words in red braces {} are sampled as latent variables, and the other words are learned with supervision. In the right panel, during training, we first select the best of top-$k$ candidates to compute the Hard-EM loss, then we combine these candidates with samples from the buffer to compute the MAPO loss. Finally, we add high-reward candidates into the buffer.

**Prompting with Question Decomposition.** Chain-of-Thought prompting (Wei et al., 2022) inserts explicit reasoning chains into prompts to help language models answer compositional and multistep questions, especially ones involving mathematical reasoning. Subsequent work such as successive (Dua et al., 2022; Zhou et al., 2023), iterative (Zelikman et al., 2022), modularized (Khot et al., 2022) or tool-based prompting (Yao et al., 2023) filter or refine the reasoning process. However, LLMs are computationally expensive and require careful prompt engineering. We show that smaller, more efficient LMs can outperform LLMs when trained to output reasoning chains.

## 3 Problem Formulation

We define the notations related to the question decomposition annotation QDMR, as well as the multistep QA training and testing setups.

### 3.1 QDMR

Formally, a QDMR for a question $q$ is a corresponding list of natural language sub-questions $\mathbf{q}^{\text{sub}} = [q_1^{\text{sub}}, q_2^{\text{sub}}, ..., q_n^{\text{sub}}]$, where answering each sub-question would lead to answering $q$ (Wolfson et al., 2020). The number of sub-questions $n = |\mathbf{q}^{\text{sub}}|$ varies for different questions $q$; if $q$ requires multistep reasoning, $n$ is usually $> 1$.

**QDMR Parser.** We assume access to a QDMR parser model $g(\cdot; \phi)$, parameterized by $\phi$, that takes in a question $q$ and generates a corresponding QDMR, as proposed by Wolfson et al. (2020). The parser is trained with the original QDMR annotation, and can predict QDMR for any new question.

### 3.2 Multistep Question Answering

A QA model takes in a question $q$ and context passage $c$, and outputs a predicted answer $\hat{a}$. We train our model on a training dataset of question-context-answer triples $(q, c, a)$. Our multistep QA model answers a question $q$ by generating and answering QDMR sub-questions one at a time.

**Training Data.** We assume that all QA training examples have a QDMR $\mathbf{q}^{\text{sub}}$ corresponding to question $q$. However, the human-annotated QDMR dataset only has gold QDMR's annotated for a small fraction of QA examples. We consider two settings: (1) only use examples with gold QDMR data for QA training ($\mathcal{D}_{\text{QDMR}}$); (2) we use the QDMR parser to generate silver QDMR for the rest of the QA training data ($\mathcal{D}_{\text{QDMR+}}$). The sub-answers $\mathbf{a}^{\text{sub}} = [a_1^{\text{sub}}, a_2^{\text{sub}}, ..., a_n^{\text{sub}}]$ to each sub-question are not included in the QDMR, as QDMR annotators only looked at the question without supporting passages. We use $\hat{\mathbf{a}}^{\text{sub}} = [\hat{a}_1^{\text{sub}}, \hat{a}_2^{\text{sub}}, ..., \hat{a}_n^{\text{sub}}]$ to denote the model predicted

sub-answers. At inference time the model is given only the question $q$ and context $c$ as input, and must generate both sub-questions and sub-answers.

# 4 Proposed Methods

We discuss how to use the QDMR annotation and the QDMR parser, as well as how to train the QA model with Hard-EM and reinforcement learning in this section. We combine question decomposition and latent variable learning to construct a generalizable multistep reasoning framework.

## 4.1 Chain-of-Questions Framework

Our model $f(;\theta)$ predicts each QDMR sub-question and its corresponding sub-answer one at a time. During training, we feed the question and context into our model $f$, as in the blue box of Fig. 2, and first ask it to predict the first sub-question, $q_1^{\text{sub}}$, along with its corresponding sub-answer, $\hat{a}_1^{\text{sub}}$. To separate the sub-question and the sub-answer, we append a special [QDMR] token to the sub-question and another special [QDMR-ANS] token to the sub-answer in the model input and output. In the second iteration, we append the gold $q_1^{\text{sub}}$ and predicted $\hat{a}_1^{\text{sub}}$ after the question and context, thereby allowing the model to predict the second sub-question and its sub-answer based on previous steps. This process is repeated until all sub-questions and sub-answers have been predicted. In the final iteration, the model outputs a [END-QDMR] tag indicating the end of iterations. We use the final sub-answer $\hat{a}_n^{\text{sub}}$ as the answer to the input question.

## 4.2 Learning with Latent Answers

We illustrate how the model $f$ learns to predict the sub-questions and sub-answers. The sub-questions are provided in the annotation, so we can simply apply supervised learning to optimize the likelihood of ground-truth sub-questions given the question, the context and any sub-answers as in Eq. (1).

$$\ell_{\text{SL}}(\theta, q, c, \hat{\mathbf{a}}^{\text{sub}})$$
$$= \sum_{j=1}^{n} -\log p_\theta(q_j^{\text{sub}} \mid q, c, q_{1:j-1}^{\text{sub}}, \hat{a}_{1:j-1}^{\text{sub}}), \quad (1)$$

where $q_{1:j}^{\text{sub}}$ denotes the set $\{q_i^{\text{sub}}\}_{i=1}^{j}$, and likewise for $a_{1:j}^{\text{sub}}$. Notice that the model is trained to generate the gold next sub-question regardless of whether its previous predicted sub-answers are correct (Bengio et al., 2015). Because the ground truth sub-answers are not provided in the training time, we regard the

intermediate sub-answers as latent variables and use Hard-EM and RL to optimize them.

**Hard-EM.** A variant of the EM algorithm, Hard-EM (Neal and Hinton, 1998) assigns the most likely values to all latent variables and maximizes the expected log-likelihood of the label based on these values. Hard-EM helps to filter spurious ways to derive the correct answer. Min et al. (2019a) use Hard-EM for weakly supervised training for multi-mention QA tasks.

Since it is computationally infeasible to enumerate all possible sets of sub-answers to find the best set, we approximately compute the best $\hat{\mathbf{a}}^{\text{sub}}$ with beam search (see Appendix A.1 for details). In particular, we pick the sequence of sub-answers $\tilde{\mathbf{a}}^{\text{sub}}$ where $\tilde{a}_n^{\text{sub}} = a$ and $\tilde{a}_{1:n-1}^{\text{sub}}$ has the highest likelihood to predict $a$:

$$\tilde{a}_{1:n-1}^{\text{sub}} = \underset{\hat{a}_{1:n-1}^{\text{sub}}}{\arg\max}\, p_\theta(a \mid q, c, \mathbf{q}^{\text{sub}}, \hat{a}_{1:n-1}^{\text{sub}}).$$

Following Hard-EM, we train the model to maximize the probability of both $\mathbf{q}^{\text{sub}}$ and $\tilde{\mathbf{a}}^{\text{sub}}$, which is equivalent to minimizing negative log likelihood:

$$\ell_{\text{H-EM}}(\theta, q, c, \mathbf{q}^{\text{sub}}, \tilde{\mathbf{a}}^{\text{sub}})$$
$$= -\log p_\theta(\mathbf{q}^{\text{sub}}, \tilde{\mathbf{a}}^{\text{sub}} \mid q, c)$$
$$= \sum_{j=1}^{n} -\log p_\theta(\tilde{a}_j^{\text{sub}} \mid q, c, q_{1:j}^{\text{sub}}, \tilde{a}_{1:j-1}^{\text{sub}})$$
$$+ \ell_{\text{SL}}(\theta, q, c, \tilde{\mathbf{a}}^{\text{sub}}). \quad (2)$$

**Reinforcement Learning.** In another perspective, we view each sub-answer as an action and the whole sub-answer set as a trajectory. By doing so, we can use reinforcement learning methods such as Memory-Augmented Policy Optimization (MAPO; Liang et al., 2018), originally designed for semantic parsing, to optimize the latent sub-answers. MAPO reduces the variance of policy gradient estimates with a memory buffer that stores high-reward trajectories (in our case, sequences of predicted sub-answers).[1]

We adapt MAPO to multistep QA as follows. While the original MAPO algorithm samples many independent trajectories using the model $f$, we instead use the trajectories from the beam, which reduces sampling time and yields better quality trajectories. During training, we maintain a replay buffer $\mathcal{B}$ of high-quality sequences of predicted sub-answers. For each example $(q, c, a)$, we choose the

---

[1]See details of the original MAPO in Appendix A.2

| Model | Method | Training Data | Testing Data (F1) DROP | DROP-CS |
|---|---|---|---|---|
| BERT | BERT-Calculator | DROP | 81.7 | 55.8 |
| | NeRd | DROP | 81.8 | 59.5 |
| GPT-3.5 * | Chain-of-Thought (4-shot) | - | 59.7 | 51.7 |
| | Standard Prompting (0-shot) | - | 40.4 | 34.1 |
| T5-B | Single-step run | DROP | 51.6 | 44.8 |
| | | DROP w/ $\mathcal{D}_{\text{QDMR+}}$ | 53.1 | 45.0 |
| | Chain-of-Questions | DROP w/ $\mathcal{D}_{\text{QDMR}}$ | 46.6 | 44.3 |
| | | DROP w/ $\mathcal{D}_{\text{QDMR+}}$ | 74.4 | 63.8 |
| | | w/o MAPO | 73.7 | 62.9 |
| | | w/o Regex | 59.1 | 56.4 |
| T5-L | Single-step run | DROP | 73.9 | 53.7 |
| | | DROP w/ $\mathcal{D}_{\text{QDMR+}}$ | 75.2 | 55.3 |
| | Chain-of-Questions | DROP w/ $\mathcal{D}_{\text{QDMR}}$ | 65.8 | 55.4 |
| | | DROP w/ $\mathcal{D}_{\text{QDMR+}}$ | **84.4** | **67.9** |
| | | w/o MAPO | 83.5 | 67.8 |
| | | w/o Regex | 78.1 | 64.4 |
| TB-T5-L | Single-step run | DROP | 81.4 | 60.1 |
| | Chain-of-Questions | DROP w/ $\mathcal{D}_{\text{QDMR+}}$ | **85.6** | **68.2** |

Table 1: F1 scores on DROP dev set and DROP-CS. **Blue** bold is the best model, **purple** bold is the second best. Chain-of-Questions outperforms other baselines on both the in-distribution dev set and the contrast set. Integrating Chain-of-Questions with TeaBReaC Pretraining further improves the performance. *We report our reproduced results of GPT-3.5, which is slightly lower than their official report (64.1 on DROP with few-shot prompting).

top 5 trajectories from the beam that are not in the replay buffer to use as out-of-memory trajectories. Next, we sample at most 5 in-memory trajectories from the replay buffer. We thus have a total of $m$ different sub-answer trajectories ($5 \leq m \leq 10$), denoted as $\{\hat{\mathbf{a}}_i^{\text{sub}}\}_{i=1}^m$, for each $(q, c)$ example. Finally, we use the F1 score of the final predicted sub-answer $\hat{a}_n^{\text{sub}}$ as the reward $R(\hat{\mathbf{a}}^{\text{sub}})$ of the trajectory, The MAPO training objective uses both sets of trajectories to derive an unbiased stratified sampling estimator of policy gradient objective:

$$
\ell_{\text{MAPO}}(\theta, q, c, \mathbf{q}^{\text{sub}}, \{\hat{\mathbf{a}}_i^{\text{sub}}\}_{i=1}^m) =
$$
$$
\sum_{\hat{\mathbf{a}}_i^{\text{sub}} \in \mathcal{B}} -\frac{r_{\mathcal{B}}}{m} R(\hat{\mathbf{a}}_i^{\text{sub}}) \log p_\theta(\hat{\mathbf{a}}_i^{\text{sub}} \mid q, c, \mathbf{q}^{\text{sub}})
$$
$$
+ \sum_{\hat{\mathbf{a}}_i^{\text{sub}} \notin \mathcal{B}} -\frac{1 - r_{\mathcal{B}}}{m} R(\hat{\mathbf{a}}_i^{\text{sub}}) \log p_\theta(\hat{\mathbf{a}}_i^{\text{sub}} \mid q, c, \mathbf{q}^{\text{sub}})
$$
$$
+ \frac{1}{m} \sum_{i=1}^m \ell_{\text{SL}}(\theta, q, c, \hat{\mathbf{a}}_i^{\text{sub}}), \tag{3}
$$

where $r_{\mathcal{B}}$ is the ratio of the number of trajectories in the buffer to the total number of sampled trajectories. As step 3 in Fig. 2, after computing the objectives, we update the replay buffer $\mathcal{B}$ with high-reward examples from the beam.

### 4.3 Chain-of-Questions Training Algorithm

As mentioned in Agarwal et al. (2019), MAPO works poorly at the beginning of training, as initially no sampled trajectories receive high reward. Hard-EM provides useful training signal at the start of training, but MAPO can help training converge to a better final model once some successful trajectories are added to the buffer. Thus, we apply a mixture weight $\lambda$ to dynamically balance Eq. (2) and (3). The overall Chain-of-Questions (CoQ) loss for a given example $(q, c, a)$ is defined as:

$$
\ell_{\text{CoQ}} = \lambda \ell_{\text{MAPO}} + (1 - \lambda) \ell_{\text{H-EM}}, \tag{4}
$$

where $\lambda$ is the proportion of examples with at least one trajectory in the replay buffer. We rely on Hard-EM at the beginning of training, but switch to MAPO as the model finds successful trajectories. Note that $\ell_{\text{SL}}$ is used in both $\ell_{\text{MAPO}}$ and $\ell_{\text{H-EM}}$.

## 5 Experiments

We present our experimental setup and show CoQ outperforms other multistep fine-tuning or prompting baselines over multiple benchmarks.

### 5.1 Experimental Details

**Datasets.** For in-distribution evaluation, we use DROP and HOTPOTQA. DROP contains 77,400 training examples and 9,536 validation examples. HOTPOTQA contains 72,928 training examples and 5,901 validation examples.[2] $\mathcal{D}_{\text{QDMR}}$ contains

---

[2]We use the two-paragraph HOTPOTQA version released by Fisch et al. (2019) for efficient training.

the question decomposition of 7,705 training examples from DROP and 6,233 training examples from HOTPOTQA. We use a T5-base (Raffel et al., 2020) QDMR parser to create $\mathcal{D}_{\text{QDMR+}}$, which has question decompositions of all training examples. We do not use QDMR examples from other datasets during training, *e.g.*, we only use the DROP examples in $\mathcal{D}_{\text{QDMR}}$ for training on DROP. For robustness evaluation, we evaluate on the contrast set DROP-CS (Gardner et al., 2020) containing 947 examples, and the adversarial set HOTPOTQA-ADV (Jiang and Bansal, 2019) containing 3,627 examples.

**Models.** For DROP, we apply CoQ with T5-Base (T5-B), T5-Large (T5-L), and TeaBReaC T5-Large (TB-T5-L) with a batch size of 16. For HOTPOTQA, the context length of examples in the 4-paragraph HOTPOTQA-ADV exceeds 512, which is the token number limit of T5. We apply CoQ with LongT5-Base (LongT5-B; Guo et al., 2022a) with a batch size of 8.

**Baselines.** We compare Chain-of-Questions to several fine-tuning methods and prompting methods. For fine-tuning methods, we compare with:

- **BERT-Calculator** (Andor et al., 2019): A unified model that uses BERT to predict programs that execute to answers on DROP.
- **NeRd** (Chen et al., 2020): A neuro-symbolic model that extends BERT-Calculator from one-step operators to two-step compositions.
- **Longformer** (Beltagy et al., 2020): A transformer encoder that combines local and global attention to process long documents.

For prompting methods, we compare with the following methods using GPT-3.5:

- **Chain-of-Thought** (Wei et al., 2022): We insert few-shot in-context examples with explicit reasoning chains to solve multistep problems. Due to the limited context window of Transformer models, we use 4-shot prompting for DROP and 2-shot prompting for HOTPOTQA. Our prompt examples consist of QDMR's paired with manually written sub-answers.
- **Standard Prompting** (Brown et al., 2020): We prompt GPT-3.5 to generate the answer without providing any in-context examples.

We engineered prompts to make these baselines as competitive as possible, as detailed in Appendix C.

**Single-step Baselines.** We compare to two single-step baselines. One is standard fine-tuning

with the given dataset, and the other is fine-tuning with QDMR-augmented contexts. For the latter, we concatenate all sub-questions from the question decomposition to $(q, c)$ as the input to the model and fine-tune it to perform single-step QA (i.e., directly generate the answer). At inference time, we first use the QDMR parser $g$ to generate the sub-questions for each question, and input them along with $(q, c)$ to the model. This baseline is similar to (Guo et al., 2022b), which uses the same QDMR parser and the same model. The only difference is they train the QDMR parser with Hard-EM.

## 5.2 Task-Specific Modifications

To match the in-distribution performance of state-of-the-art systems, we make task-specific modifications for DROP and HOTPOTQA.

**Modifications for DROP.** Smaller language models such as T5-B and T5-L struggle with numerical operations. Since DROP focuses on numerical reasoning, analogous to how BERT-Calculator help the model to do arithmetic, we add a regular expression matching module that can handle basic numerical operations. We note that only the last sub-question of a DROP example may require numerical operation. Hence, we add regular expression matching and the "[REGEX]" tag in the last step.

For each example, we take the last sub-question generated by the model $f$, parse it to a functional program based on the keyword matching and execute it. If the parsing and execution process are both successful, we put the "[REGEX]" token in front of the numerical execution result and input them together into the answer generation process. Else, we keep the answer generation process unchanged as in Fig. 2.

**Modification for HOTPOTQA.** Predicting the supporting facts is an auxiliary task of HOTPOTQA used in many models (Groeneveld et al., 2020; Beltagy et al., 2020). Following the same input format as Longformer, we add the supporting fact (SF) prediction and the span prediction (SP) tasks in the encoder as auxiliary tasks. We use a two-layer feedforward network for supporting fact prediction, and one-layer classification head for span prediction. We perform these two tasks at each run of model $f$ and add the two cross-entropy losses to $\ell_{CoQ}$.

| Model | Method | Training Data | Testing Data (F1) | |
|---|---|---|---|---|
| | | | HOTPOTQA | HOTPOTQA-ADV |
| Longformer | Longformer | HOTPOTQA | **85.6** | 77.7 |
| GPT-3.5 | Chain-of-Thought (2-shot) | - | 66.8 | 58.9 |
| | Standard Prompting (0-shot) | - | 69.7 | 57.1 |
| LongT5-B | Single-step run | HOTPOTQA | 85.4 | 77.5 |
| | | HOTPOTQA w/ $\mathcal{D}_{QDMR+}$ | 85.2 | 78.2 |
| | Chain-of-Questions | HOTPOTQA w/ $\mathcal{D}_{QDMR}$ | 74.7 | 72.8 |
| | | HOTPOTQA w/ $\mathcal{D}_{QDMR+}$ | **85.1** | **83.2** |
| | | w/o MAPO | 83.0 | **82.0** |
| | | w/o SF & SP | 78.1 | 76.5 |

Table 2: F1 scores on HOTPOTQA dev set and HOTPOTQA-ADV. **Blue** bold is the best model, **purple** bold is the second best. Chain-of-Questions matches the performance of Longformer on the in-distribution dev set, and outperforms all baselines on the adversarial set.

## 5.3 Results

We show results for DROP and DROP-CS in Table 1, and results for HOTPOTQA and HOTPOTQA-ADV in Table 2. The results indicate the effectiveness and robustness of Chain-of-Questions.

**Chain-of-Questions outperforms other baselines.** In Table 1, CoQ w/ $\mathcal{D}_{QDMR+}$ on DROP outperforms all baselines on F1 by 2.6% in-distribution and 7.8% on the contrast set. Moreover, the T5-L version is 3.5% better than recently released GPT-4 F1 score on the DROP (OpenAI, 2023). The model can be further improved by initializing with TeaBReaC Pretraining. Similarly, in Table 2, CoQ w/ $\mathcal{D}_{QDMR+}$ on HOTPOTQA is on-par in-distribution with Longformer and 5.5% higher on the adversarial set. On both datasets, CoQ has a smaller performance gap between the in-distribution dev set and the robustness evaluation set compared with the baselines, indicating its strong robustness.[3]

**Chain-of-Thought prompting is weaker than fine-tuning methods on multistep QA.** On both DROP and HOTPOTQA, prompting methods have lower F1 than fine-tuning methods, which indicates the difficulty of prompting large language models to do multistep QA. Moreover, Chain-of-Thought is 2.9% worse than zero-shot Standard Prompting on HOTPOTQA F1 score. We find it difficult to design prompts for GPT-3.5 that lead to clean and concise answers under the Chain-of-Thought setup. HOTPOTQA-ADV benefits from question decomposition, which suggests Standard Prompting may take reasoning shortcuts on HOTPOTQA examples.

**MAPO works better on HOTPOTQA than DROP.** Using MAPO in addition to Hard-EM, as opposed to Hard-EM alone, leads to larger gains in F1 on HOTPOTQA (+[1.2-2.5]%) than on DROP (+[0.1-0.9]%). As the sub-answers should be spans in the context for both DROP and HOTPOTQA, our hypothesis is that the span prediction task provides an inductive bias for actions to focus on spans in the context, which makes the model more likely to generate correct answers. The context of 2-paragraph HOTPOTQA is shorter than DROP, which makes the action space smaller.

**Task-specific modifications are necessary.** Both the regular expression module in DROP and the auxiliary tasks (SF & SP) in HOTPOTQA improves F1 by more than 5% on the in-distribution dev sets and 3% on the robustness evaluation sets in Table 1 and 2. This shows that task-specific modifications are an important part of Chain-of-Questions.

**Chain-of-Questions can generalize to benchmarks with no QDMR annotation.** To test if our framework is still effective when annotated QDMR's are not available for the training dataset, we conduct additional experiments where we omit the QDMR annotations for one dataset, then run CoQ on that dataset using only QDMR generated by a QDMR parser trained on other datasets.

Recall that in our main DROP experiments, $\mathcal{D}_{QDMR+}$ includes both human-annotated gold QDMR and silver QDMR generated by a QDMR parser trained with the QDMR annotation of DROP. Instead, we now train a QDMR parser with only the QDMR annotation of COMPLEXWEBQUESTIONS (Talmor and Berant, 2018) and HOTPOTQA. Then, we use that QDMR parser to generate bronze QDMR augmentation for the whole DROP dataset, and use these to run CoQ. In this

---

[3] The one exception is the prompting baselines on DROP, which are 30% lower on DROP and 20% lower on DROP-CS than CoQ.

| Train Data | DROP | DROP-CS |
|---|---|---|
| DROP | | |
| w/ $\mathcal{D}_{QDMR}$ | 46.6 | 44.3 |
| w/ bronze QDMR | 69.1 | 59.8 |
| w/ $\mathcal{D}_{QDMR+}$ | 74.4 | 63.8 |

| Train Data | HOTPOTQA | HOTPOTQA-ADV |
|---|---|---|
| HOTPOTQA | | |
| w/ $\mathcal{D}_{QDMR}$ | 74.7 | 72.8 |
| w/ bronze QDMR | 84.9 | 81.5 |
| w/ $\mathcal{D}_{QDMR+}$ | 85.5 | 83.2 |

Table 3: F1 scores of Chain-of-Questions models on DROP dev set, DROP-CS, HOTPOTQA dev set and HOTPOTQA-ADV set, trained with different QDMR data. By assuming DROP and HOTPOTQA has no QDMR annotation, CoQ models trained on bronze QDMR drops 4-5% F1 on DROP and DROP-CS, 1-2% F1 on HOTPOTQA and HOTPOTQA-ADV, while they are still much better than the gold-only QDMR training, as well as the single-run baselines.

way, we consider what would happen if no annotated QDMR was available for DROP. Similarly, for HOTPOTQA, we train the QDMR parser with the QDMR annotation of COMPLEXWEBQUESTIONS and DROP, and generate bronze QDMR for the HOTPOTQA dataset.

Compared to training on $\mathcal{D}_{QDMR+}$, CoQ training on bronze QDMR results in decreases of 4-5% F1 on DROP and DROP-CS and 1-2% F1 on HOTPOTQA and HOTPOTQA-ADV. Nonetheless, these F1 scores are still much better than the single-run baselines on the contrast and adversarial sets. These experiments show that CoQ can be effective even on benchmarks without QDMR annotations. Training on bronze QDMR is also much more effective than training only on gold $\mathcal{D}_{QDMR}$, as shown in Table 3. This suggests that having a large amount of training data greatly helps CoQ, even if that data is lower quality; future work could explore using more unannotated data to further improve performance.

### 5.4 Qualitative Analysis on QDMR

We conduct qualitative analysis on QDMR to check its quality and effectiveness.

**How good are the generated sub-questions and sub-answers?** We list three multistep QA examples in Table 4 to show quality of generated QDMR sub-questions and sub-answers.[4] The generated

---

[4]We choose the examples based on following the criteria (1) the example must be a multistep QA (2) we manually check around 3 examples per dataset satisfying (1) and select one

sub-questions from DROP (1st) and HOTPOTQA (3rd) examples are correct decompositions. In the DROP-CS (2nd) example, the generated QDMR is also a valid decomposition, although answering the first two generated sub-questions require additional reasoning steps. Note that the QDMR is annotated by only looking at the questions (Wolfson et al., 2020), and the QDMR sub-question generation in Chain-of-Questions is trained by supervised learning. The QDMR annotation may cause the sub-question generation to focus only on the question, and thus generate sub-questions that require multiple reasoning steps from the passage, which are hard for the model to answer.

The final answer of the question is generated in the first sub-answer in the HOTPOTQA (3rd) example, which is not a correct answer to the first sub-question. The model may use reasoning shortcuts to generate the final answer as sub-answers, which is beneficial to generate the final answer at the last step, but not answering the sub-question.

**How do the generated sub-questions help the QA model?** We hypothesize generated sub-questions helps the model return the right answer for the right reason, and we try to detect this by looking at whether it can correctly identify supporting facts in HOTPOTQA-ADV. For each question, the model predicts two supporting facts in the context using the encoder output embedding. We observe that the accuracy of supporting fact prediction improves as we incorporate question decomposition in the input.

For example, given a question *"What government position was held by the woman who portrayed Corliss Archer in the film Kiss and Tell?"*, the ground truth supporting facts are two sentences: *"Kiss and Tell is ... Shirley Temple as Corliss Archer."* and *"As an adult, she ... served as Chief of Protocol of the United States."*, from two different paragraphs.

In the first run, the input to the encoder is only the question and context; the model predicts the first supporting fact correctly, while it takes a sentence *"As an adult, she ... served as Chief of treaty of the United States."* from an adversarial paragraph, which refers to a different person, as the second supporting fact. However, in the second run, when the input to the encoder contains the first sub-question and sub-answer ("[QDMR] return

---

based on how well the context and the relevant information can be visualized.

| Question | Context | Ground-truth QDMR | Generated QDMR & Answers |
|---|---|---|---|
| How many years after Pegu fell did the king die? (DROP) | After the fall of Pegu **in December 1599** ... but the king died during the campaign on **3 March 1606**. | 1. return year that Pegu fell; 2. return year that the king did die; 3. return difference of #2 and #1. | 1. return the year when Pegu fell: 1599; 2. return the year when the king died: 1606; 3. return difference of 1606 and 1599: **7**. |
| Which cities did they go before Manchester and Derby? (DROP-CS) | The Jacobite army ... enteblue England on 8 November ... **Two days later they reached Carlisle** ... **On 26 November, they reached Preston** ... followed by Manchester on the 28th November ... enteblue Derby on 4 December ... | 1. return the date they went to Manchester; 2. return the date they went to Derby; 3. return the cities they went before #1; 4. return the cities they went before #2; 5. return the cities in both #3 and #4. | 1. return cities they go before Manchester: Preston; 2. return cities they go before Derby: Manchester; 3. return the cities in both Preston and Manchester: **Preston**. |
| Which airline partners with Gogo Inflight Internet and opened in 1984? (HOTPOTQA) | Gogo Inflight ... **17 airlines partner with Gogo** to provide in-flight WiFi, **including** ... **Virgin Atlantic**, a trading name of ... **was established in 1984** ... | 1. return airline that partners with Gogo Inflight Internet; 2. return #1 that opened in 1984. | 1. return which airline partners with Gogo Inflight Internet: Virgin Atlantic; 2. return Virgin Atlantic open in 1984: **Virgin Atlantic**. |

Table 4: Qualitative analysis of model-generated reasoning chains. We use **black** bold to mark relevant information in the context, **blue** bold to mark correct final answers, and **red** bold to mark wrong final answers. We find (1) most generated sub-questions are correct; (2) a generated sub-question by CoQ models can contain multiple reasoning steps; (3) the model may use reasoning shortcuts to generate the final answer as sub-answers for early sub-questions.

the woman who portrayed Corliss Archer in the film Kiss and Tell [QDMR-ANS] Shirley Temple"), the model selects the second sentence from the paragraph about Shirley Temple, thus predicting both supporting facts correctly.

For all the 2-hop questions in HOTPOTQA-ADV, the supporting fact prediction accuracy is 75.6% during the first run, and rises to 83.2% during the second run. This accuracy increase shows the effectiveness of generating sub-questions step-by-step, which helps the model gradually filter out close but irrelevant context.

## 6 Discussion

We present Chain-of-Questions (CoQ), a robust sub-question generation and answering framework that shows strong performance on DROP and HOTPOTQA. CoQ uses a combination of Hard-EM and MAPO for training, effectively optimizing the latent variables associated with sub-answers of intermediate questions.

We envision multiple directions for future work. CoQ requires supervision from QDMR; other families of RL methods we did not explore may be used to reduce our reliance on this supervision, and instead allow the model to learn appropriate decompositions from scratch. On the other hand, we could also explore using different question decompositions, such as ones generated by LLMs like GPT-3.5. Either approach could help us extend CoQ to other multistep reasoning datasets with no QDMR annotation. Similar to DROP, FINQA (Chen et al., 2021) consists of numerical reasoning questions over financial data. In a similar format as HOTPOTQA, MUSIQUE (Trivedi et al., 2022a) contains 3-hop and 4-hop retrieval questions. ROPES (Lin

et al., 2019) requires complex multistep reasoning between a background context paragraph and situation context paragraph. We could either train models on these datasets if we can eliminate our reliance on QDMR data, or test whether models trained with CoQ can transfer well to these other datasets.

## Limitations

Due to GPU resource constraints, we were unable to scale up our method to larger models such as T5-3B. However, the smaller models we were able to experiment with already show good performance. Similar, for computational efficiency, we did not try other more advanced on-policy reinforcement algorithms, but we find that MAPO already yields good improvement on F1.

Chain-of-Questions still requires task-specific modifications for different multistep QA benchmarks—we did not find out a good way to build a universal model that is highly effective on all datasets. UnifiedQA (Khashabi et al., 2020b) constructs a unified model for multiple QA benchmarks, but their largest model (T5-11B) still has poor performance on DROP, again suggesting the importance of dataset-specific modifications.

Finally, the best version of Chain-of-Questions requires QDMR annotation during training, which is only available for some datasets. In order to be independent of the QDMR annotation, we show some generalization of CoQ by assuming no QDMR annotation on DROP and training on bronze question decompositions generated by QDMR parser. CoQ could also be tested on other datasets without QDMR annotation to evaluate its transferability, by having the QDMR parser for bronze annota-

tions. On the other hand, the method can be further explored to work with question decompositions generated by LLMs.

## Acknowledgements

We thank Qinyuan Ye, Ameya Godbole, Kristina Toutanova, Jonathan Berant and the members of USC NLP Group for their valuable feedback. This work was supported in part by an Open Philanthropy research grant and a Google Research Scholar Award.

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

## A  Chain-of-Questions Algorithm Details

We list omitted details of Hard-EM and MAPO in the Chain-of-Questions algorithm.

### A.1  Sub-Answer Sampling Strategy

Given $\mathbf{q}^{\text{sub}}$ and $\hat{\mathbf{a}}^{\text{sub}}$, we compute the likelihood to predict the sequence of $\mathbf{q}^{\text{sub}}$ and $\hat{\mathbf{a}}^{\text{sub}}$ by:

$$
p_\theta(\mathbf{q}^{\text{sub}}, \hat{\mathbf{a}}^{\text{sub}} \mid q, c) =
$$
$$
\prod_{j=1}^{n} p_\theta(\hat{a}_j^{\text{sub}} \mid q, c, q_{1:j}^{\text{sub}}, \hat{a}_{1:j-1}^{\text{sub}})
$$
$$
\prod_{j=1}^{n} p_\theta(q_j^{\text{sub}} \mid q, c, q_{1:j-1}^{\text{sub}}, \hat{a}_{1:j-1}^{\text{sub}}) \quad (5)
$$

We use beam search to sample the sub-answers and keep a beam of size $k$. In each iteration, we expand each beam with $b$ different answers and select the top-$k$ likelihood answers to construct a new beam following Eq. (5). We choose $k = 25$ and $b = 5$ in our experiments. Thus, we have 5 different candidates for $\hat{a}_1^{\text{sub}}$ in the first run, 25 different candidates for $(\hat{a}_1^{\text{sub}}, \hat{a}_2^{\text{sub}})$ after the second run, and in general 25 candidates for $\hat{a}_{1:j}^{\text{sub}}$ for the $j$-th run for $j > 2$.

The challenge is to ensure the sampling will provide some correct sub-answers. However, notice that roughly 10% of annotated QDMR's across DROP and HOTPOTQA are single-step decomposition (i.e., $n = 1$). Our hypothesis is the model may to learn to do single-step QA from these examples, which will also help it learn to produce meaningful sub-answers to sub-questions that come from multistep QDMR's.

### A.2  Difference from Original MAPO

Given a policy model $\pi(; \theta)$, and a replay buffer $\mathcal{B}$, the original MAPO objective is:

$$
\mathcal{O}_{\text{MAPO}} = r_\mathcal{B} \mathbb{E}_{\hat{\mathbf{a}}^{\text{sub}} \sim \pi_\theta^+(\hat{\mathbf{a}}^{\text{sub}})} R(\hat{\mathbf{a}}^{\text{sub}})
$$
$$
+ (1 - r_\mathcal{B}) \mathbb{E}_{\hat{\mathbf{a}}^{\text{sub}} \sim \pi_\theta^-(\hat{\mathbf{a}}^{\text{sub}})} R(\hat{\mathbf{a}}^{\text{sub}})
$$

where $R(\hat{\mathbf{a}}^{\text{sub}})$ denotes the reward of the trajectory, and $r_\mathcal{B}$ is the ratio of the number of trajectories in the buffer to the total number of sampled trajectories, used to derive an unbiased stratified sampling estimator of the gradient. $\pi_\theta^+(\hat{\mathbf{a}}^{\text{sub}})$ and $\pi_\theta^-(\hat{\mathbf{a}}^{\text{sub}})$ are the normalized probability distribution inside and outside the buffer, defined as:

$$
\pi_\theta^+(\hat{\mathbf{a}}^{\text{sub}}) = \pi_\theta(\hat{\mathbf{a}}^{\text{sub}})/r_\mathcal{B} \cdot \mathbb{1}[\hat{\mathbf{a}}^{\text{sub}} \in \mathcal{B}]
$$
$$
\pi_\theta^-(\hat{\mathbf{a}}^{\text{sub}}) = \pi_\theta(\hat{\mathbf{a}}^{\text{sub}})/(1 - r_\mathcal{B}) \cdot \mathbb{1}[\hat{\mathbf{a}}^{\text{sub}} \notin \mathcal{B}]
$$

Notice that sampling a new trajectory is expensive in our scenario, especially with a large model. So instead of use samples from a policy model, we instead use the trajectories from the beam, which reduces sampling time and yields better quality trajectories.

On the other hand, MAPO stores trajectories with rewards greater than 0 in $\mathcal{B}$. However, wrong answers with some corrects words could also get a F1 score greater than 0 with our reward function. Hence, we select $R(\hat{\mathbf{a}}^{\text{sub}}) > 0.8$ and store these trajectories in the replay buffer $\mathcal{B}$.

## B  Implementation Details

We list the implementation details and hyperparameter choice as follows.

### B.1  Datasets

Jiang and Bansal (2019) constructed HOTPOTQA-ADV by generating up to 8 adversarial paragraphs for a given HOTPOTQA example. Because we trained on the 2-paragraph HOTPOTQA, we generate a 4-paragraph version of HOTPOTQA-ADV to reduce length generalization. We take the adversarial set from Jiang and Bansal (2019), filter the examples with at least 2 adversarial paragraphs, and select their intersection with the validation set of 2-paragraph HOTPOTQA. We randomly choose the 2 adversarial paragraphs, and randomly order them with the 2 supporting paragraphs.

### B.2  DROP Regular Expression

During the execution time, we detect the output [END-QDMR] token for the last sub-question.

We search for specific keywords in the last sub-question and match them with numerical operations, *e.g.*, we match "higher" to max and "less" to min. The keyword matching perfectly matches the ground truth QDMR annotation to numerical operations. We use 7 operators in total: max, min, sum, diff, mul, div, or.

We take the last sub-question generated by the model $f$, parse it to a functional program based on the keyword matching and execute it. If the parsing and execution process are both successful, we put the "[REGEX]" token in front of the numerical execution result and input them together into the answer generation process. For example, if the last sub-question generated is "[QDMR] return the largest of 4 and 3 [END-QDMR]", we will input "[QDMR] return the largest of 4 and

| Model | DROP | DROP-CS |
|---|---|---|
| TB-T5-L | 85.6±0.5 | 68.2±0.8 |
| Model | HOTPOTQA | HOTPOTQA-ADV |
| LongT5-B | 85.1±0.3 | 83.0±0.5 |

Table 5: The mean F1 score and the standard deviation on the evaluation benchmarks of the best CoQ models. The results are based on three different runs.

```
3 [END-QDMR] [REGEX] 4" into the decoder for
```
answer generation, in both training and inference time.

### B.3 Hyperparameters

In training DROP, we train a total of 50k iterations with a batch size of 8 for T5-B and a batch size of 4 for T5-L. One run of DROP training with T5-B on one NVIDIA V100 GPU takes 40 hours. One run of DROP training with T5-L on one NVIDIA A100 GPU takes 60 hours.

In training HOTPOTQA, we train a total of 30k iterations with a batch size of 16 for LongT5-B. One run of HOTPOTQA training with LongT5-B on one NVIDIA V100 GPU takes 30 hours.

In all training, we use the AdamW optimizer with a weight decay of 0.01. The learning rate is set as $1e^{-5}$. We evaluate the model every 500 steps and set an early-stopping criterion on the validation F1 score, with max patience of 3.

### B.4 Standard Deviation

We report the mean F1 score and the standard deviation over three runs of the best CoQ models in Table 5, which are the TB-T5-L model for DROP and the LongT5-B model for HOTPOTQA.

## C Prompt Engineering

We show the effort we made on prompt engineering to get the best possible performance on DROP and HOTPOTQA using GPT-3.5.

### C.1 Zero-shot Prompting

The zero-shot prompt we used to evaluate the model is

```
"Please answer the question after
reading the context. You should start
with the thinking and reasoning steps,
such as retrieving relevant content from
the context and solving the question
step-by-step. You should give the final
answer in the format of 'The answer is
```

```
_'. The final answer must be short and
concise. Don't repeat the question in the
final answer. \n\n"
```

The last two sentences help the model to generate clean and concise answers. Without these two sentences in the prompt, the zero-shot F1 score will drop by more than 10% in DROP and HOTPOTQA benchmarks. Without the thinking step-by-step sentence, the F1 performance will drop more (-8%) on DROP but less (-2%) on HOTPOTQA, indicating HOTPOTQA has more validation questions containing reasoning shortcuts.

However, even with these constraints and explicit format provided in the prompt, the GPT-3.5 model is still hard to generate the precise answer in the format we want. For example, apart from 'The answer is', GPT-3.5 generates different strings in front of the answer, such as 'The final answer is', 'Final answer is', 'Answer:'. During the answer parsing, we consider all different variations of the pre-answer string to parse for the answer.

### C.2 Few-shot Prompting

The few-shot prompt we used to evaluate the model is

```
"[INSTRUCTION] [EXAMPLE] [REASONING]
[ANSWER] [EXAMPLE] [REASONING] [ANSWER]
... [EXAMPLE]"
```
where

```
[INSTRUCTION] = "We provide 4 examples
for answering the question given the
context, by thinking step-by-step. Please
answer the last question in the same
format. \n\n"
```

[EXAMPLE] = "question: $q$, context: $c$ \n",

```
[REASONING] = "question decomposition:
```
$q_1^{sub}$: $a_1^{sub}$; ...; $q_n^{sub}$: $a_n^{sub}$. \n",

```
[ANSWER] = "The answer is a \n\n"
```
We tried different variations for each tag as follows and evaluated on 100 random examples to choose the best prompt over their combinations.

```
[INSTRUCTION] = "Please answer the
question after reading the context. You
should start with the thinking and
reasoning steps, such as retrieving
relevant content from the context and
solving the question step-by-step. We
provide 4 examples for answering the
question given the context, by thinking
step-by-step. You should give the final
```

answer in the format of 'The answer is
_'. The final answer must be short and
concise. Don't repeat the question in the
final answer. \n\n"

  [REASONING] = "Let's think step-by-step:
$q_1^{\mathsf{sub}}$: $a_1^{\mathsf{sub}}$; ...; $q_n^{\mathsf{sub}}$: $a_n^{\mathsf{sub}}$. \n",

  [ANSWER] = "Answer: $a$ \n\n"

The results on 100 random validation examples
of DROP and HOTPOTQA both show that shorter
and more concise instruction with a more formal
reasoning format achieves better F1 score.

In terms of the cost, we did not optimize for the
best few-shot examples for validation data. Instead,
we selected 5 random set of few-shot examples
and chose the best set over them by F1 score over
100 validation examples. For each few-shot exam-
ples set, we wrote the sub-answers for the QDMR
annotations and inserted these sub-answers into
the QDMR question if they are referenced. The an-
swer parsing mechanism is the same as in zero-shot
prompting.

## D   Random Qualitative Examples

We use `numpy.random.choice` to select 2 random
examples from DROP and 2 random examples from
HotpotQA. Unlike in Table 4, we omit the manually
written ground truth QDMR for the dev set and use
the ground truth answer instead.

| Question | Context | Generated QDMR & Answers | GT Answer |
|---|---|---|---|
| How many of the personnel were not military specialists? (DROP) | The total number of active military personnel .... In May 2016, Armed Forces had 16,019 members, of which **14,506 were active military personnel** and 1,513 civil servants. Of the 14,506 active military personnel, 3,183 were officers, ... **520 military specialists**, ... | 1. return number of personnel: 14,506 were active military personnel. 2. return number of military specialists: 520 military specialists. 3. return the differences of 14,506 were active military personnel and 520 military specialists: 13,986. | 13,986 |
| How many more people are there than housing units? (DROP) | As of the census of 2000, there were **49,129 people**, ... **There were 21,779 housing units** at an average density of 39 per square mile (15/km$^2$) ... | 1. return number of people: 49,129 people. 2. return number of housing unit: 21,779. 3. return the differences of 49,129 people and 21,779: 27,350. | 27,350 |
| This cabin built in 1925 is located in a park encompasses over 1 million acre and includes parts of two what? (HOTPOTQA) | Lower Park Creek Patrol Cabin. **The Lower Park Creek Patrol Cabin in Glacier National Park is a rustic backcountry log cabin. Built in 1925, the cabin has**... Glacier National Park is a national park ... **The park encompasses over 1 million acre and includes parts of two mountain ranges (sub-ranges of the Rocky Mountains)**, over 130 named lakes, ... | 1. return park this cabin built in 1925: Glacier National Park. 2. return Glacier National Park encompasses over 1 million acre and includes parts of two what: mountain ranges. | mountain ranges |
| Bandit was built in 1988 by which Japanese amusement ride company that built roller coasters, giant wheels, carousels, flumes, dark rides, sky cycles and other amusement rides? (HOTPOTQA) | Bandit (Yomiuriland)... **Built in 1988 by the TOGO company**... TOGO. **TOGO (株式会社トゴ , Kabushiki-gaisha Tōgo ) was a Japanese amusement ride company that built roller coasters, giant wheels, carousels, flumes, dark rides, sky cycles and other amusement rides**. | 1. return the Japanese amusement ride company that built roller coasters, giant wheels, carousels, flumes, dark rides, sky cycles and other amusement rides: TOGO (株式会社トゴ, Kabushiki-gaisha Tōgo ). 2. return TOGO (株式会社トゴ, Kabushiki-gaisha Tōgo ) built bandit in 1988: TOGO. | TOGO company |

Table 6: Qualitative analysis of model-generated reasoning chains. We use **black** bold to mark relevant information in the context.