# OpenReview forum: "Chain-of-Questions Training with Latent Answers for Robust Multistep Question Answering"
_EMNLP/2023/Conference — EMNLP 2023 Main_

### Official Review · Reviewer_dfZy · 2023-08-02

**Soundness:** 4

**Excitement:**

3: Ambivalent: It has merits (e.g., it reports state-of-the-art results, the idea is nice), but there are key weaknesses (e.g., it describes incremental work), and it can significantly benefit from another round of revision. However, I won't object to accepting it if my co-reviewers champion it.

**Paper Topic And Main Contributions:**

This paper introduces the Chain-of-Questions framework, designed to enhance the robustness of multistep question answering by generating and answering sub-questions. While human-annotated question decomposition meaning representation (QDMR) provides supervision for sub-questions, the absence of annotated sub-question answers poses a technical challenge. To address this, the paper presents a novel approach that treats sub-answers as latent variables, employing a dynamic mixture of Hard-EM and MAPO for inference. The effectiveness and robustness of Chain-of-Questions are demonstrated through impressive performance gains, outperforming strong neuro-symbolic methods by 9.0 F1 on a DROP contrast set and surpassing GPT-3.5 by 24.3 F1 on a HotpotQA adversarial set. Overall, this paper introduces the innovative Chain-of-Questions framework for robust multistep question answering. The framework showcases impressive performance gains, surpassing neuro-symbolic methods and GPT-3.5 on various evaluation sets. The paper addresses the absence of annotated sub-question answers through a novel technical solution, further enhancing the framework's robustness. Although there are limitations in terms of evaluation measures and real-world application, the paper's contributions and advancements make it a valuable addition to the field of multistep question answering.



**Reasons To Accept:**


The Chain-of-Questions framework presents a novel approach to robust multistep question answering, introducing the generation and answering of sub-questions as a key component. This innovative framework sets the paper apart from existing methods in the field.

The results showcase significant performance gains, with Chain-of-Questions outperforming strong neuro-symbolic methods by 9.0 F1 on a DROP contrast set. Additionally, the framework surpasses GPT-3.5 by a notable margin of 24.3 F1 on a HotpotQA adversarial set, highlighting its effectiveness in challenging scenarios.

Addressing the technical challenge of lacking annotated answers to sub-questions, the paper proposes a solution that treats sub-answers as latent variables and employs a dynamic mixture of Hard-EM and MAPO for inference. This technical innovation enhances the applicability and robustness of the framework.

By focusing on multistep question answering and providing impressive performance gains across different evaluation sets, the paper demonstrates the robustness and generalizability of the Chain-of-Questions framework.

The paper compares the framework's performance against strong neuro-symbolic methods and GPT-3.5, providing a comprehensive evaluation that highlights the superiority of Chain-of-Questions.


**Reasons To Reject:**

While the paper highlights impressive F1 gains on specific evaluation sets, a broader range of evaluation metrics would provide a more comprehensive understanding of the framework's performance.

While the results demonstrate superior performance on contrast and adversarial sets, the absence of evaluation on real-world datasets may limit the practical application and impact of the proposed framework.

The technical solution presented in the paper, involving a dynamic mixture of Hard-EM and MAPO, may introduce complexity during implementation. Further details or simplifications could assist researchers in adopting the proposed approach more easily.



**Reproducibility:**

3: Could reproduce the results with some difficulty. The settings of parameters are underspecified or subjectively determined; the training/evaluation data are not widely available.

**Reviewer Confidence:**

3: Pretty sure, but there's a chance I missed something. Although I have a good feel for this area in general, I did not carefully check the paper's details, e.g., the math, experimental design, or novelty.

---

> ### Author Rebuttal · Authors · 2023-08-29
>
> We thank the reviewer for their feedback and are glad they acknowledge Chain-of-Questions is an innovative framework apart from existing methods in the field, as well as its effectiveness in challenging scenarios.
>
> **Q1: While the paper highlights impressive F1 gains on specific evaluation sets, a broader range of evaluation metrics would provide a more comprehensive understanding of the framework's performance.**
>
> A1: Thanks for your suggestions. For completeness, we will add EM score in the camera-ready version, the overall result trends and takeaways are the same as F1 scores.
>
> **Q2: While the results demonstrate superior performance on contrast and adversarial sets, the absence of evaluation on real-world datasets may limit the practical application and impact of the proposed framework.**
>
> A2: We would like to point out many previous works on DROP and HotpotQA [1,2,3,4] do not evaluate on real-world datasets. Also, DROP focuses on arithmetic multi-hop reasoning and HotpotQA focuses on entity-centric multi-hop reasoning. They are quite different benchmarks, which already shows the generality of the CoQ approach.
>
> Moreover, we view generalizing the CoQ method to arbitrary QA dataset as an orthogonal contribution to developing it in the first place.
>
>
>
>
> **Q3: Further details or simplifications could assist researchers in adopting the proposed approach more easily.**
>
> A3: We will release the code after the anonymity period ends.
>
> [1] Teaching Broad Reasoning Skills for Multi-Step QA by Generating Hard Contexts (Trivedi et al., EMNLP 2022)
>
> [2] OPERA: Operation-Pivoted Discrete Reasoning over Text (Zhou et al., NAACL 2022)
>
> [3] From Easy to Hard: Two-Stage Selector and Reader for Multi-Hop Question Answering (Li et al. ICASSP 2023)
>
> [4] Is Graph Structure Necessary for Multi-hop Question Answering? (Shao et al., EMNLP 2020)

---

### Official Review · Reviewer_UiWR · 2023-08-03

**Typos Grammar Style And Presentation Improvements:** Table 1 caption
**Soundness:** 3

**Excitement:**

4: Strong: This paper deepens the understanding of some phenomenon or lowers the barriers to an existing research direction.

**Missing References:**

[1] Kunlong Chen, Weidi Xu, Xingyi Cheng, Zou Xiaochuan, Yuyu Zhang, Le Song, Taifeng Wang,
Yuan Qi, and Wei Chu. Question directed graph attention network for numerical reasoning over
text. arXiv preprint arXiv:2009.07448, 2020.

**Paper Topic And Main Contributions:**

This paper proposes the Chain-of-Questions framework that generates and answers subquestions for answering complex questions that require multiple steps of reasoning. The model is trained with QDMR but treats sub-answers as latent variables and infers them using a novel dynamic mixture of Hard-EM and MAPO. The Chain-of-Questions framework outperforms selected neuro-symbolic methods and GPT-3.5 on DROP, HotpotQA, and their respective contrast and adversarial datasets.

**Questions For The Authors:**

A. Is the performance of the BERT-based models reported in Table 1 based on BERT-base or BERT-large?

B. Are the results reported in Tables 1 and 2 the maximum, mean, etc., or just from a single run?

C. What are the existing SOTAs of DROP-CS and HOTPOTQA-ADV?

**Reasons To Accept:**

- A novel dynamic mixture of Hard-EM and MAPO for learning with latent answers.
- CoQ is showing complementary benefiths with TeaBReaC.
- Solid improvements over single-step run demonstrating the effectiveness of CoQ.

**Reasons To Reject:**

- Baseline models are not strong enough. The performance of GPT3.5 based models may be underestimated in this paper as the GPT-4 Technical Report reported 64.1 F1 on DROP for 3-shot GPT-3.5. Other strong models like QDGAT[1], leading models at https://hotpotqa.github.io/ are not considered.
- The "fast improvement at the beginning and better final convergence" of the dynamic mixture of Hard-EM and MAPO lacks supporting evidence in the paper.
- The generalization of the proposed method may be challenging due to the prerequisite of QDMR annotation and task-specific preprocessing.

**Reproducibility:**

3: Could reproduce the results with some difficulty. The settings of parameters are underspecified or subjectively determined; the training/evaluation data are not widely available.

**Reviewer Confidence:**

4: Quite sure. I tried to check the important points carefully. It's unlikely, though conceivable, that I missed something that should affect my ratings.

---

> ### Author Rebuttal · Authors · 2023-08-29
>
> We thank the reviewer for their comments and are pleased they find CoQ is novel, effective, and shows solid improvements over baseline models.
>
> **Q1: Baseline models are not strong enough.**
>
> **Q1a: The performance of GPT3.5 based models may be underestimated in this paper as the GPT-4 Technical Report reported 64.1 F1 on DROP for 3-shot GPT-3.5.**
>
> A1a: Thanks for pointing this out. We noticed this, and we had tried our best with prompting in Appendix C. The GPT-4 report did not specify their prompt, and we are not able to reproduce their results on DROP. On the other hand, the 64.1 number in the GPT-4 report is still worse than CoQ and other baselines.
>
> **Q1b: DROP baselines, other strong models like QDGAT**
>
> A1b: We tried to run the QDGAT model before to evaluate on DROP-CS, which is our main focus. However, their codebase is not runnable (same for OPERA, others on the top have no code released). Specifically, the QDGAT codebase mentioned “run parse_xml.py” in their README.md, but that file is not in their codebase. They also don't allow posting issues. Notice that TeaBreaC [1], the SotA on DROP-CS as far as we know, also did not compare with these methods.
>
> **Q1c: HotpotQA baselines, leading models at the leaderboard are not considered.**
>
> A1c: As mentioned in the footnote of page 5, we use the two-paragraph version released by MRQA 2019 (Fisch et al, 2019) for efficient training. Most methods on the leaderboard with code released focus on selecting two relevant paragraphs out of ten, such as R3, FE2H [3, 4]. These methods will almost be the same as Longformer on the two-paragraph setup.
>
> **Q2: The "fast improvement at the beginning and better final convergence" of the dynamic mixture of Hard-EM and MAPO lacks supporting evidence in the paper.**
>
> A2: Thank you for pointing it out. We will add a convergence figure similar to Fig.2 in MAPO [5], which shows Hard-EM converges faster for a few thousand steps and then is outperformed by Hard-EM+MAPO in the middle.
>
> **Q3: The generalization of the proposed method may be challenging due to the prerequisite of QDMR annotation and task-specific preprocessing.**
>
> A3: We agree that the generalization to arbitrary multistep QA dataset is challenging, and we view generalizing to other un-annotated benchmarks is a research question of its own.
> On the other hand, we tried the following setups to test the generalization abilities on other un-annotated benchmarks using QDMR parser.
>
> Notice that $\mathcal{D}_\mathrm{QDMR+}$ is augmented with silver QDMRs generated by a QDMR parser trained with the QDMR annotation of DROP. Instead, we train a QDMR parser with the QDMR annotation of CWP and HotpotQA and generate the bronze QDMR augmentation for DROP. In this way, if we consider DROP as an un-annotated benchmarks, we could still use DROP w/ bronze QDMR to train.
>
> *T5-B Chain-of-Questions results on DROP and DROP-CS*
>
> Train Data | DROP | DROP-CS
> :---|:---:|:---:
> DROP w/ gold QDMR ($\mathcal{D}_\mathrm{QDMR}$) | 46.6 | 44.3
> DROP w/ bronze QDMR | 69.1 | 59.8
> DROP w/ gold+silver QDMR ($\mathcal{D}_\mathrm{QDMR+}$) | 74.4 | 63.8
>
> CoQ training with bronze QDMR drops 4-5% F1 on DROP and DROP-CS while it is still much better than the single-run baselines (53.1/45.0), suggesting the potential of CoQ to generalize to un-annotated benchmarks. We will include these results in the camera-ready.
>
>
> **Q4: Is the performance of the BERT-based models reported in Table 1 based on BERT-base or BERT-large?**
>
> A4: BERT-Calculator and NeRd both use BERT-base.
>
> **Q5: Are the results reported in Tables 1 and 2 the maximum, mean, etc., or just from a single run?**
>
> A5: Single-run. As mentioned in Appendix B.3, experiments on RL take long to train. But we will report the average and std of 3 runs in the camera-ready version.
>
> **Q6: What are the existing SotAs of DROP-CS and HotpotQA-ADV?**
>
> A6: As far as I know, the existing SotA of DROP-CS is TeaBreaC-3B [1], which achieves F1 score 66.8, we use TeaBreaC-L with CoQ and outperform them (68.5).
>
> For existing SotA of HotpotQA-ADV, note that we mention the setup we are using is 4 paragraphs (as explained in Appendix B.1) instead of 10 paragraphs. Moreover, instead of releasing the dataset, the HotpotQA-ADV paper [2] released a method of creating the dataset. The numbers in their paper are on a dataset created with randomness. In their codebase, they write “in order to recreate the adversarial training data we used in the paper, randomly sample 40% of the adversarial training data generated using this code and combine with the original HotpotQA training set.” Thus, the numbers are not comparable.
>
> [1] Teaching Broad Reasoning Skills for Multi-Step QA by Generating Hard Contexts (Trivedi et al., EMNLP 2022)
>
> [2] Avoiding Reasoning Shortcuts: Adversarial Evaluation, Training, and Model Development for Multi-Hop QA (Jiang and Bansal, ACL 2019)
>
> [3] From Easy to Hard: Two-Stage Selector and Reader for Multi-Hop Question Answering (Li et al. ICASSP 2023)
>
> [4] Rethinking Label Smoothing on Multi-hop Question Answering (Yin et al., 2022)
>
> [5] Memory Augmented Policy Optimization for Program Synthesis and Semantic Parsing (Liang et al., 2019)

---

### Official Review · Reviewer_7ULL · 2023-08-07

**Soundness:** 3

**Excitement:**

4: Strong: This paper deepens the understanding of some phenomenon or lowers the barriers to an existing research direction.

**Paper Topic And Main Contributions:**

The authors propose a Question Answering (QA) method for multi-step QA, where the question is first decomposed into sub-questions, which are then solved prior to being used to answer to the original question.

The authors train a question decomposition model trained on a smaller set of annotated data of decomposing questions, and use this model to generate decompositions for the entire training set. These decomposed questions are then answered by the model, successively, and used to answer the next decomposition until arriving at the answer. The author propose to use HardEM and the RL method MAPO to optimize their model, slowly transitioning from HardEM loss during training to attain better final convergence of RL with the superior initial stability of HardEM.

The authors test their method on HotPotQA and DROP datasets, and show strong performance improvements in both datasets. The generated questions and answers

**Questions For The Authors:**

1. Were the sample questions and answers shown in Table 3 hand-picked, or random? If hand-picked, how does a random set of generated answers and questions look like? If these were representative illustrative examples, perhaps a table of randomly selected examples should also be added to the appendix.

**Reasons To Accept:**

1. The proposed method is intuitive, clear, and results in performance improvements across multiple datasets.
1. Combines multiple prior directions such as question generation, question decomposition, and multi-step QA to yield a well-motivated method.
1. The method gives good performance even with simple HardEM based training if a practitioner does not want additional complexity/instability of RL.
1. The transfer of loss from HardEM to MAPO is well motivated and results in performance improvements

**Reasons To Reject:**

Generalization of the authors' method to other datasets (as mentioned in line 583-585, 610-613) is critical to make the paper stronger.
As most datasets lack QDMR decomposition, atleast simple transferring of their method using silver predicted data on other datasets should be studied, to make the method more generally applicable.


**Reproducibility:**

4: Could mostly reproduce the results, but there may be some variation because of sample variance or minor variations in their interpretation of the protocol or method.

**Reviewer Confidence:**

4: Quite sure. I tried to check the important points carefully. It's unlikely, though conceivable, that I missed something that should affect my ratings.

**Typos Grammar Style And Presentation Improvements:**

Line 970 "model is still hard to generate" should be reworded to be grammatically correct.

---

> ### Author Rebuttal · Authors · 2023-08-29
>
> We thank the reviewer for their feedback. We are pleased they find CoQ is intuitive, clear, well-motivated, and they acknowledge performance improvements across multiple datasets.
>
> **Q1: Generalization of the authors' method to other datasets (as mentioned in line 583-585, 610-613) is critical to make the paper stronger.**
>
> A1: Thank you for the comments. We tried the following setups to test the generalization abilities on other un-annotated benchmarks using QDMR parser.
>
> Notice that $\mathcal{D}_\mathrm{QDMR+}$ is augmented with silver QDMRs generated by a QDMR parser trained with the QDMR annotation of DROP. Instead, we train a QDMR parser with the QDMR annotation of CWP and HotpotQA and generate the bronze QDMR augmentation for DROP. In this way, if we consider DROP as an un-annotated benchmarks, we could still use DROP w/ bronze QDMR to train.
>
> *T5-B Chain-of-Questions results on DROP and DROP-CS*
>
> Train Data | DROP | DROP-CS
> :---|:---:|:---:
> DROP w/ gold QDMR ($\mathcal{D}_\mathrm{QDMR}$) | 46.6 | 44.3
> DROP w/ bronze QDMR | 69.1 | 59.8
> DROP w/ gold+silver QDMR ($\mathcal{D}_\mathrm{QDMR+}$) | 74.4 | 63.8
>
> CoQ training with bronze QDMR drops 4-5% F1 on DROP and DROP-CS while it is still much better than the single-run baselines (53.1/45.0), suggesting the potential of CoQ to generalize to un-annotated benchmarks. We view generalizing to other un-annotated benchmarks as an orthogonal research direction, but we will include these results in the camera-ready.
>
> **Q2: Were the sample questions and answers shown in Table 3 hand-picked, or random? If hand-picked, how does a random set of generated answers and questions look like?**
>
> A2: Thank you for the suggestion. They are hand-picked based on the criteria (1) the example must be a multistep QA (2) we look at ~3 examples per dataset satisfying (1) and select one based on how well the context and the relevant information (bold) can be visualized.
>
> A random set of generated answers and questions (we use numpy.random.choice) will look similar as Table 3. A table of 2 random examples from DROP and 2 random examples from HotpotQA is like below. We omit the manually written ground truth QDMR for the dev set and use the ground truth answer instead. We will add a longer table of randomly selected examples in the Appendix in the camera-ready.
>
> Question | Context | Generated QDMR & Answers | GT Answer
> :---|:---:|:---:|---:
> How many of the personnel were not military specialists? (DROP) | The total number of active military personnel .... In May 2016, Armed Forces had 16,019 members, of which **14,506 were active military personnel** and 1,513 civil servants. Of the 14,506 active military personnel, 3,183 were officers, ... **520 military specialists**, ... | 1. return number of personnel: 14,506 were active military personnel. 2. return number of military specialists: 520 military specialists. 3. return the differences of 14,506 were active military personnel and 520 military specialists: 13,986. | 13,986
> How many more people are there than housing units? (DROP) | As of the census of 2000, there were **49,129 people**, … **There were 21,779 housing units** at an average density of 39 per square mile (15/km2)... | 1. return number of people: 49,129 people. 2. return number of housing unit: 21,779. 3. return the differences of 49,129 people and 21,779: 27,350. | 27,350
> This cabin built in 1925 is located in a park encompasses over 1 million acre and includes parts of two what? (HotpotQA) | Lower Park Creek Patrol Cabin. **The Lower Park Creek Patrol Cabin in Glacier National Park is a rustic backcountry log cabin.  Built in 1925, the cabin has**... Glacier National Park is a national park ...  **The park encompasses over 1 million acre and includes parts of two mountain ranges (sub-ranges of the Rocky Mountains)**, over 130 named lakes, ... | 1. return park this cabin built in 1925: Glacier National Park. 2. return Glacier National Park encompasses over 1 million acre and includes parts of two what: mountain ranges.  | mountain ranges
> Bandit was built in 1988 by which Japanese amusement ride company that built roller coasters, giant wheels, carousels, flumes, dark rides, sky cycles and other amusement rides? (HotpotQA) | Bandit (Yomiuriland)...  **Built in 1988 by the TOGO company**... TOGO. **TOGO (株式会社トーゴ , Kabushiki-gaisha Tōgo ) was a Japanese amusement ride company that built roller coasters, giant wheels, carousels, flumes, dark rides, sky cycles and other amusement rides**. | 1. return the Japanese amusement ride company that built roller coasters, giant wheels, carousels, flumes, dark rides, sky cycles and other amusement rides: TOGO (株式会社トーゴ , Kabushiki-gaisha Tōgo ). 2. return TOGO (株式会社トーゴ , Kabushiki-gaisha Tōgo ) built bandit in 1988: TODO. | TOGO company

---

### Meta-Review · Area_Chair_tMG8 · 2023-09-18

**Recommendation:** 4

**Metareview:**

The authors propose to split questions that require multiple steps into finer grained questions, answering them and then combining this to generate the final answer. This is done by training a model over QDMR parsings. As QDMR annotation does not have intermediate answers, the authors use hard EM to train the model to answer the questions.

Reviewers overall like this approach and find the methodology convincing.

On the other hand, the reviewers think that the experimental setup in the paper is limiting as the authors does not evaluate on an eval set for which there is no training data.
I agree with this -- this is indeed limiting as QDMR parsings are only available for a limited amount of dataset, and hard expensive to produce. In general the setup would be that there is no training set for the target domain so it is important to test the generalization of the proposed method.

---

### Decision · Program_Chairs · 2023-10-07

**Decision:**

Accept-Main

**Comment:**

The authors propose to split questions that require multiple steps into finer grained questions, answering them and then combining this to generate the final answer. This is done by training a model over QDMR parsings. As QDMR annotation does not have intermediate answers, the authors use hard EM to train the model to answer the questions.

Reviewers overall like this approach and find the methodology convincing.

On the other hand, the reviewers think that the experimental setup in the paper is limiting as the authors does not evaluate on an eval set for which there is no training data.
I agree with this -- this is indeed limiting as QDMR parsings are only available for a limited amount of dataset, and hard expensive to produce. In general the setup would be that there is no training set for the target domain so it is important to test the generalization of the proposed method.